# Chocolate intake and muscle pain sensation: A randomized experimental study

**Alexandra Hajati[1], Mario Brondani[2], Lina Angerstig[1], Victoria Klein[1], Linda Liljeblad[1], Essam Ahmed Al-Moraissi[3], Sofia Louca Jounger[1], Bruna Brondani[2], Nikolaos Christidis[1] ***

**1** Division of Oral Diagnostics and Rehabilitation, Department of Dental Medicine, Karolinska Institutet, and Scandinavian Center for Orofacial Neurosciences, Huddinge, Sweden, **2** Division of Dental Public Health, Department of Oral Health Sciences, Faculty of Dentistry, The University of British Columbia, Vancouver, Canada, **3** Department of Oral and Maxillofacial Surgery, Faculty of Dentistry, Thamar University, Thamar, Yemen

* Nikolaos.Christidis@ki.se

## Abstract

### Background

Chocolate, as a cocoa-derived product rich in flavanols, has been used for medical and anti-inflammatory purposes. Therefore, the aim of this study was to investigate if the ingestion of different percentages of cocoa products affects the experimentally induced pain caused by intramuscular hypertonic saline injections in the masseter muscle of healthy men and women.

### Methods

This experimental randomized, double-blind, and controlled study included 15 young, healthy, and pain-free men and 15 age-matched women and involved three visits with at least a 1-week washout. Pain was induced twice at each visit with intramuscular injections of 0.2 mL hypertonic saline (5%), before and after intake of one of the different chocolate types: white (30% cocoa content), milk (34% cocoa content), and dark (70% cocoa content). Pain duration, pain area, peak pain, and pressure pain threshold (PPT) were assessed every fifth minute after each injection, up until 30 min after the initial injection. Descriptive and inferential statistics were performed using IBM® SPSS (Version 27); significance level was set to $p < 0.05$.

### Results

This study showed that intake of chocolate, no matter the type, reduced the induced pain intensity significantly more than no intake of chocolate ($p < 0.05$, Tukey test). There were no differences between the chocolate types. Further, men showed a significantly greater pain reduction than women after intake of white chocolate ($p < 0.05$, Tukey test). No other differences between pain characteristics or sexes were revealed.

**Data Availability Statement:** All relevant data are within the manuscript and its Supporting Information files.

**Funding:** The author(s) received no specific funding for this work.

**Competing interests:** The authors have declared that no competing interests exist.

## Conclusion

Intake of chocolate before a painful stimulus had a pain-reducing effect no matter the cocoa concentration. The results indicate that perhaps it is not the cocoa concentration (e.g., flavanols) alone that explains the positive effect on pain, but likely a combination of preference and taste-experience. Another possible explanation could be the composition of the chocolate, i.e. the concentration of the other ingredients such as sugar, soy, and vanilla.

## Introduction

Pain is considered a global health issue since it not only influences quality of life for one-fifth of the world's adult population, but also causes massive costs for patients, the health care system, and society [1, 2]. Luckily, most times pain will be transient, as it is associated with a lesion or disease that will heal. The word pain comes from a Greek word known as poine, which means penalty. In modern times, pain is described as "an unpleasant sensory and emotional experience associated with actual or potential tissue damage or described in terms of such damage" [3].

According to its temporal and aetiological categorization, pain can be classified as acute or chronic, and as nociceptive, neuropathic, nociplastic, idiopathic (e.g. pain of unknown origin), or mixed pain [4, 5]. Acute nociceptive pain is a sensation which arises when noxious peripheral stimuli activate the free nerve endings of nociceptive pathways. This stimulus does not need to induce tissue damage but has the intensity to reach the pain thresholds in order to elicit a pain sensation. Hence, acute pain is also a protective mechanism that helps prevent further injury by generating a reflex withdrawal [6, 7]. Chronic pain, on the other hand, is defined as pain that persists after healing and continues for at least 3 months [8].

To understand clinical pain, experimental pain models are utilized with healthy and homogenous participants, without confounding factors such as medications, age differences, illnesses, or external factors such as weather, ongoing wars. Furthermore, they can be performed in a standardized and controlled setting, with distinct start and end-points allowing quantitative assessments [9]. The use of hypertonic saline as the substance to chemically induce pain of muscular origin in the orofacial region in particular has been deemed valid [10–13]. Its acute nature and pronounced sensation of deep and diffuse pain, along with pain referral, is believed to mimic the sensation of patients with orofacial pain of muscular origin [11–17].

A common orofacial pain is associated with the temporomandibular disorder (TMD), described as all pain conditions affecting the masticatory muscles (and associated structures) and/or the temporomandibular joints [18]. Myalgia, i.e., pain in the jaw muscles, is the most common type of TMD, and leads to decreased mouth opening, pain when chewing or moving the jaw, soreness, and headaches [18]. Pain associated with TMD has a prevalence ranging from approximately 10 to 20% and is 1.5 to 2 times higher in women than men [13, 18–21]. The mechanisms behind why men are less predisposed to having TMD than women are still not clear [22].

It is suggested that TMD myalgia has a multifactorial aetiology due to complex interactions between biological, psychological, social, and environmental factors [23, 24]. When it comes to the biological factors, which are of interest here, some studies have suggested that micro-inflammation is involved in the development of TMD myalgia, where algogenic substances activate or sensitize nociceptive-free nerve endings, thereby eliciting a pain sensation [25–28].

One of these algogenic substances is the neurotransmitter serotonin, a target for many psycho-pharmaceutical drugs [29, 30]. Serotonin, which is found in blood platelets and in the central nervous system (CNS) [29], is released due to tissue damage or ischemia, as well as during inflammation; however, it has also been shown to regulate mood [31, 32]. Serotonin concentrations in patients with chronic myalgia have been shown to be significantly higher than those without [33].

Cocoa is believed to be linked to serotonin through tryptophan, a precursor of serotonin and an essential amino acid found in this widely consumed dietary product [34]. Cocoa has been used for medical and anti-inflammatory purposes throughout history, and it has been shown that cocoa-derived products rich in flavanols can reduce inflammation [35, 36].

It has also been found that a cocoa-enriched diet inhibits neurogenic inflammatory pain in rats, which implies the possible use of cocoa as an alternative therapy for pain control in humans [37]. There are indications that the type of chocolate (e.g., percentage of cocoa solids) plays an important role regarding its effect on sensory experiences [34]. However, the preference for a certain type of chocolate (e.g., sugar content, texture, and aroma) could also have an effect on pain based on a person's psychological state or mood [34, 38–40] given that the composition of chocolate varies considerably [41].

Taken together, the aim of this study was to investigate if the ingestion of products with different percentages of cocoa affects the experimentally induced pain caused by intramuscular injections of hypertonic saline into the masseter muscle in healthy men and women. The hypothesis of the project was that the higher the cocoa content of the chocolate consumed, the less pain is experienced.

## Materials and methods

The experiment took place between March 1 and December 20, 2020, at a research lab in the Department of Dental Medicine, Karolinska Institutet, Huddinge, Sweden. Ethics approval was obtained from the Swedish Ethical Review Authority (Dnr: 2019/05785), and the project followed the principles for medical research according to the declaration of Helsinki, as well as Good Clinical Practice (GCP) guidelines. Furthermore, the study was registered on Clinical-Trials.gov (Identifier: NCT05378984). The participants received both written and verbal information and gave their verbal and written consent.

### Participants

The minimum sample size was estimated based on a normally distributed standard deviation of 30% [9, 29], a significance level (α) of 0.05 and a power (β) of 99%. In turn, 13 pairs of healthy participants were warranted. Considering the risk of dropouts, 15 men and 15 women were included and attended all three sessions without any dropouts, as shown in the CON-SORT Flow Diagram (Fig 1). The power to detect a significant effect within the 15 pairs was also guaranteed at >0.999 (99%) using an effect (f) of 3, an error probability (α) of 0.05 and a correlation among the repeated 6 measures of 0.5.

Inclusion criteria for participating was men and women aged between 18–40 years in good general health. Exclusion criteria eliminated those with any pain-related diagnosis of TMD in the orofacial region, with headaches, systemic muscular or joint diseases (fibromyalgia or rheumatoid arthritis), whiplash-associated disorders, neurological disorders, psychiatric disorders, or allergies to any of the substances used. Recruitment of participants was done mainly through information to students at the Odontology Department at Karolinska Institutet in Huddinge, Sweden. Further information was given about the study, both in writing and verbally, to participants before the study began.

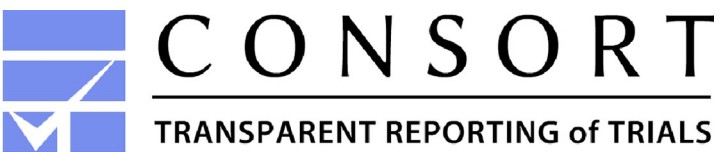

## CONSORT 2010 Flow Diagram

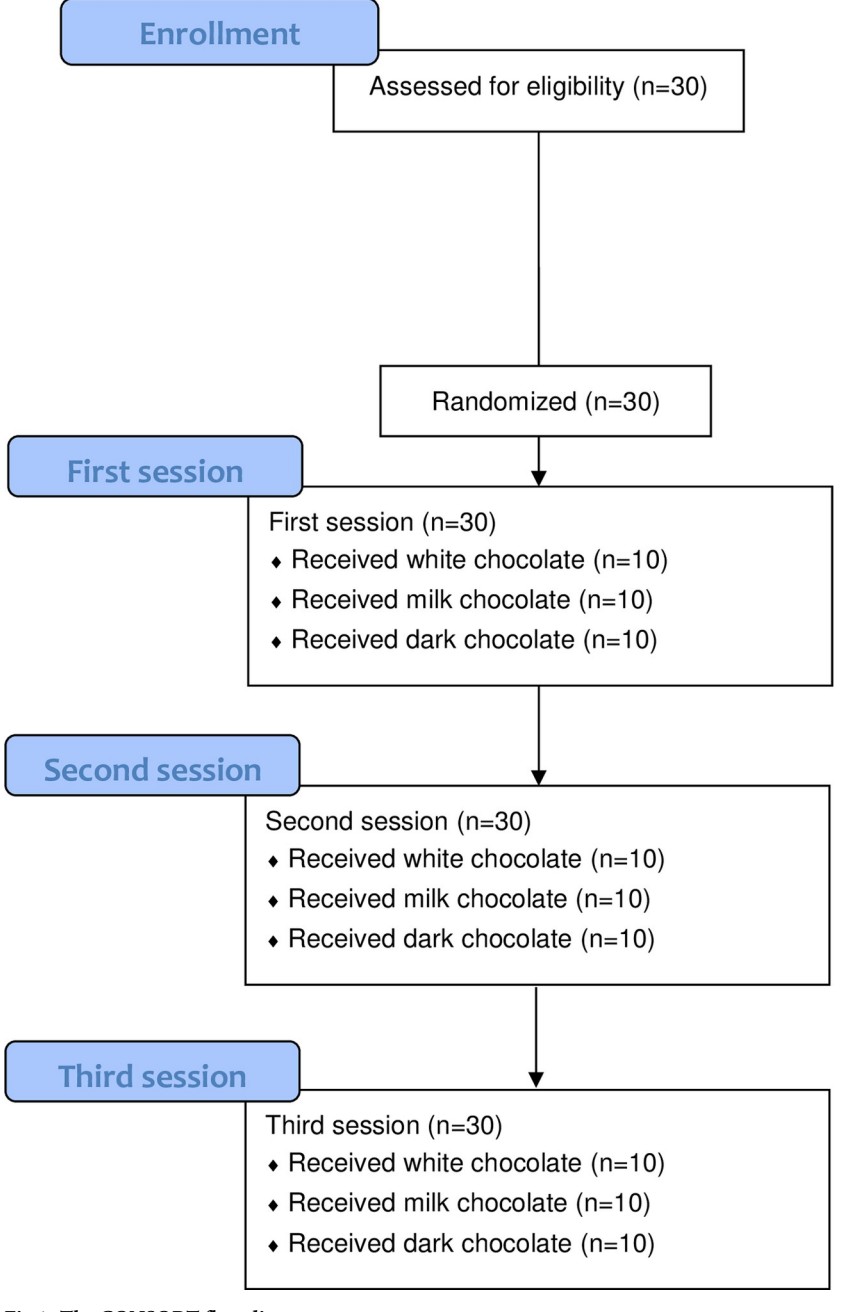

**Fig 1. The CONSORT flow diagram.**

## Study design

The study was designed as a randomized, controlled, and double-blind study. All participants completed questionnaires regarding their psychosocial status at the first visit prior to inclusion, and these included depression, somatization, anxiety, pain catastrophizing and stress (Axis II of the Diagnostic Criteria for Temporomandibular Disorders (DC/TMD)) [18]. The following questionnaires were used to assess self-reported emotional functioning:

- The *Patient Health Questionnaire* (PHQ-9): to assess self-reported depression symptoms and mental disorders including 9 items (each scored 0–3) based on the criteria for mental disorders in the DSM-IV diagnostic criteria for depressive disorders. The overall score is calculated into different levels of severity depending on the points: normal (0–4 points), mild (5–9 points), moderate (10–14 points), moderately severe (15–19 points), and severe (20–27 points) [42].

- The *Generalized Anxiety Disorder* screener (GAD-7): to assess self-reported anxiety, consisting of 7 items (each graded 0–3) for assessing degree of anxiety related to pain. The total score according to severity is divided into: normal (0–4 points), mild (5–9 points), moderate (10–14 points), and severe (15–27 points) [43].

- The *Patient Health Questionnaire* (PHQ-15): to assess self-reported somatization and non-specific physical symptoms, consisting of 15 items (each graded 0–2) and divided from normal to severe nonspecific physical symptoms: normal (0–4 points), mild (5–9 points), moderate (10–14 points), and severe (15–30 points) [44].

- The *Pain Catastrophizing Scale* (PCS-13): to assess self-reported feelings of rumination and magnification regarding pain, including 13 items (each scored 0–4) depending on the extent. The total score according to the risk of pain catastrophizing is divided into: normal (0–19 points), risk of pain catastrophizing (20–29 points), and high risk of pain catastrophizing (≥30 points) [45].

- The *Perceived Stress Scale* (PSS-10): to assess the severity of stress including 10 items (each scored 0–4) depending on frequency of thoughts and feelings regarding stress during the last 30 days. The overall score according to the level of stress: normal level of stress (0–12 points), moderate level of stress (13–20 points), and severe level of stress (21–40 points) [46].

Prior to inclusion, all participants underwent examination of the orofacial region according to the DC/TMD [18] by a blinded examiner (AH–for all men; and VK–for all women).

**1. Induction of experimental pain.**  During each experimental session (~ 1 hour), acute pain was induced by intramuscular injections (0.2 ml) of sterile hypertonic saline (58.5 mg/ml) into the most prominent point of the right side masseter muscle (assessed during contraction) by a non-blinded examiner (LA–for all men; and LL–for all women). Before injection, the skin was cleaned with a swab containing isopropyl alcohol (70%). To ensure intramuscular injection, a cannula of 19*0.4 mm was used and inserted perpendicular to the skin-surface covering the masseter muscle to a depth of 15 mm, as previously described by Christidis *et al.* [11]. Immediately after injection, assessments of pain characteristics and pressure pain thresholds (PPT) were performed. After 5 minutes pain had disappeared, but assessments continued for 30 minutes.

After the first 30 minutes of the experiment, including injection and assessments, one random piece of chocolate (3.6 g) was provided to the participant and 5 minutes later a second injection of hypertonic saline was given in the exact same manner as the first injection. All assessments were repeated in the same manner, as shown in Fig 2. Thus, each participant

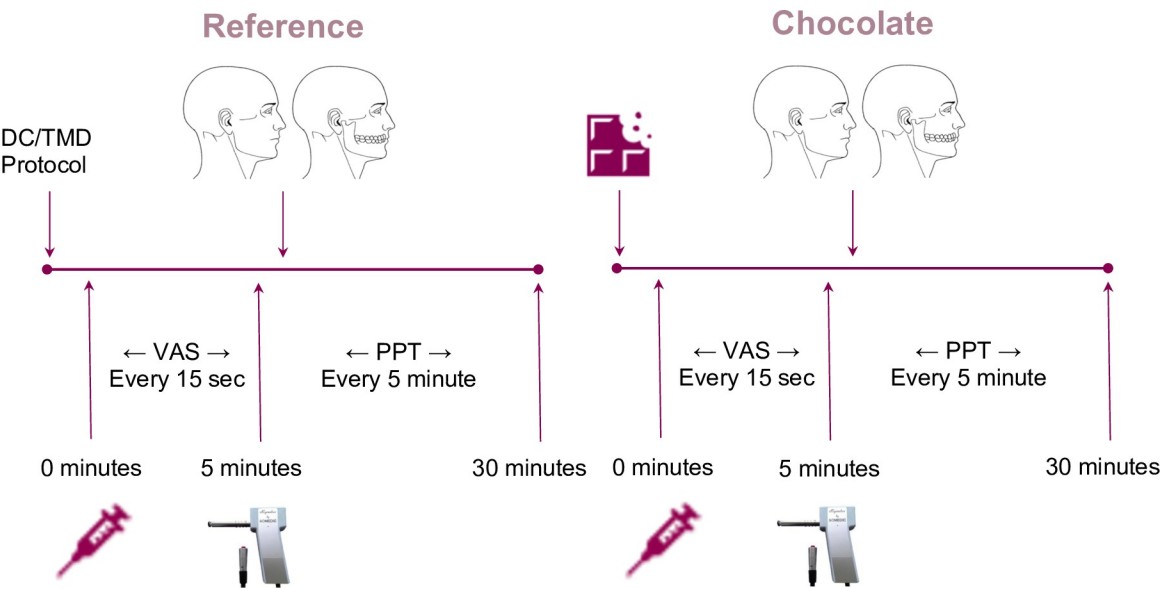

**Fig 2. Study design flowchart.** At 0 minutes hypertonic saline is injected into the masseter muscle, every fifth minute PPT is assessed for 30 minutes, then this is repeated after intake of either white, milk, or dark chocolate, in healthy age-matched men and women (15 of each). DC/TMD = Diagnostic Criteria for Temporomandibular Disorders. VAS = Visual Analogue Scale (0–100). PPT = Pressure Pain Threshold.

served as their own control (pre- or post-chocolate intake). This procedure has also been explained in detail in a study by Christidis *et al.* [11].

**2. Intake of chocolate.** The chocolate given to the participants contained a different concentration of cocoa: 30% (white), 34% (milk), and 70% (dark), distributed in a randomized and double-blinded order. The participants were given the chocolate with their eyes closed. The non-blinded examiner (LA, LL) placed the chocolate in the participant's mouth in each experiment, while the blinded examiner (AH, VK) did all the assessments. To ascertain that there was no possible carry-over effect, the different types of chocolate were given at three different visits to all participants with at least 1 week of washout between visits. The participants were asked to tell which of the three chocolates they preferred once they completed all three sessions.

To randomize the order in which the chocolates were given to each participant, an internet-based site was used (www.randomization.com; Seed 19525). The randomization was done in five blocks of six participants each, by a researcher who did not participate in the data collection.

The chocolates used in this study were packaged at Gem Chocolates®, Vancouver, Canada, using wafers from Belcolade manufacture in Belgium. They were all made uniform in shape and placed in homogeneous bags, making them blinded for participants and examiners. Each bag contained a single type of chocolate of a given concentration of cocoa. Each type of chocolate was made with a variation in the concentration of cocoa mass (cocoa nibs + cocoa butter), sugar, emulsifier, natural vanilla flavoring, and other products. Based on the US Food & Drug Administration (FDA), the chocolates were:

Dark chocolate: content of 70% cocoa (cocoa beans: 67.495%, cocoa butter: 3.2%, sugar: 28.3%, whole milk powder: 0.5%, sunflower lecithin: 0.4% and e476: 0.1% [emulsifiers], and natural vanilla extract: 0.005%),

Milk chocolate: content of 34% cocoa (cocoa beans: 9.8%, cocoa butter: 25.9%, sugar: 42.9%, whole milk powder: 21%, sunflower lecithin: 0.4% [emulsifier], and natural vanilla flavor: 0.04%),

White chocolate: content of 30% cocoa (cocoa butter: 30%, sugar: 43.16%, whole milk powder: 21.2%, dried skimmed milk: 5%, lecithin: 0.6% [emulsifier], natural vanilla flavor: 0.04%) [47].

The manufacturer also advises that their product may contain traces of soy. Of note, cocoa content comes from both the cocoa butter and the cocoa nibs that together make the cocoa mass (that is, the same 70% dark chocolate can have different percentages of cocoa nibs and cocoa butter and still be 70%).

**3. Assessment of pain variables.** The participants were requested to grade their perceived pain intensity continuously on a 0 to 100 mm Visual Analogue Scale (VAS), with 0 representing "no pain" and 100 representing "worst imaginable pain". Pain intensity was marked as participants were prompted to do so every fifteenth second for 5 minutes (20 grades in total). At the end of the last grading, 5 minutes after injection, the participants were asked to mark the maximal subjective induced pain spread using a chart drawing with two lateral views of the head, one extraoral and one intraoral.

**4. Assessment of pressure pain threshold (PPT).** PPT was assessed using an electronic pressure algometer (Somedic Sales AB, Hörby, Sweden) with a 1 cm$^2$ probe tip covered in 1 mm rubber on the skin surface of the masseter muscle. The pressure was applied at the most prominent point of the masseter muscle, coinciding with the site of the saline injection. Using a horizontal angle, the pressure was increased by the blinded examiner at a controlled rate of 30kPa/s. When pressure merged into pain, the subject was instructed to press a signal button. The PPTs were recorded one time every fifth minute after the injection for a total of 6 measurements within 30 minutes, i.e. 5, 10, 15, 20, 25, and 30 minutes after injection. The recording of PPT was only done once at each time point to reduce the risk of sensitization by repeated pressure stimuli. A standardized point was also chosen as the reference point on the left or right index fingertip on which the PPT was also assessed every fifth minute.

## Data analyses

Statistical analyses were conducted on pain intensity, peak pain, pain duration, and pain area. Face charts marking the pain area were scanned using a Ricoh MP C6004ex printer with 300 dpi resolution, followed by using Adobe Photoshop CC2019 (Adobe Systems Incorporated USA) to mark the encirclements on the drawings and to express them in arbitrary units (au).

## Statistical analyses

Statistical analysis was performed using SPSS 28 (SPSS Inc. Chicago, IL, USA). To test the normality of the data, the Shapiro-Wilk's test was used. Mean and standard deviation (SD) were used for descriptive statistics when data was normally distributed, while median and interquartile range (IQR) were used for non-normally distributed data. Moreover, when data was normally distributed, parametric statistical methods were used, while non-parametric statistics were used for data that was not normally distributed.

The Kruskall-Wallis one-way analysis of variance (ANOVA) on ranks was used to test if there were any differences regarding background factors among the groups and different chocolate types.

The Shapiro-Wilk's test indicated that data regarding pain intensity and PPT were normally distributed. Based on this, two-way repeated measures (RM) ANOVA was used to analyze

differences over time, with chocolate type as the independent factor and time as the repeated factor. When the RM ANOVA indicated that there was a significant time difference, the Tukey test, for multiple comparison versus a control group (baseline), was used as a post-hoc test, as well as to test differences between chocolate types and interactions at different time points. Before statistical analysis of pain intensity and PPT, the values were normalized to baseline. Thus, the pre- and post-chocolate difference, i.e. the relative changes (%) were used in the statistical analyses.

Data regarding peak pain intensity, pain duration, and pain area from pain drawings were not normally distributed, not even after log transformation. Therefore, these data were analyzed using non-parametric methods. The Friedman RM ANOVA on ranks was used to test for differences between chocolate types. When a significant difference was indicated, the Dunn's method for multiple comparisons versus a control group (baseline) was used as a post-hoc test. Since there were three different types of chocolate this was repeated three times, thus a Bonferroni correction was used resulting in a significance level of $p < 0.017$ for this analysis. To test the effect of intake of low cocoa-content (30–34%) chocolate against high cocoa-content chocolate (70%) and to test for sex differences, the Mann-Whitney U-test was used.

The significance level was set to $p < 0.05$ for all tests.

## Results

### Demographic data

Fifteen healthy and pain-free men with a mean (SD) age of 24.1 (3.4) years and 15 age-matched healthy and pain-free women with a mean (SD) age of 25.1 (2.6) years participated in this study.

The Kruskall-Wallis one-way ANOVA on ranks did not show any differences in psychosocial state between the different types of chocolate (Table 1). The scores for PHQ-9, PSS-10, PCS, GAD-7, and PHQ-15 were also similar.

Milk chocolate was considered the favorite kind of chocolate for 27 out of 30 (14 men and 13 women) participants on a scale from 1 to 3, with a mean score of 2.73 points. White chocolate had a mean score of 2.40 points, while dark chocolate was the least favored, with a mean score of 1.87 points.

### Pain intensity over time

**White chocolate.** The two-way RM ANOVA showed a significant time effect (F = 92.498; $p < 0.001$), a significant difference with or without intake of white chocolate (F = 4.433; $p = 0.044$), and a significant interaction between time and intake of white chocolate (F = 1.782; $p = 0.017$). The post-hoc test showed that the experimentally induced pain intensity after intake of white chocolate was significantly lower than without intake 105–210 s after induction of pain ($p < 0.05$, Tukey test), as shown in Fig 3.

In men, the two-way RM ANOVA showed a significant time effect (F = 38.136; $p < 0.001$), a significant difference with or without intake of white chocolate (F = 11.240; $p = 0.005$), and a significant interaction between time and intake of white chocolate (F = 2.576; $p < 0.001$). The post-hoc test showed that the experimentally induced pain intensity after intake of white chocolate was significantly lower than without intake 60-240s after induction of pain ($p < 0.05$, Tukey test), as shown in Fig 3.

In women, the two-way RM ANOVA showed a significant time effect (F = 56.698; $p < 0.001$), but no difference with or without intake of white chocolate (F = 0.0483; $p = 0.829$), and no interaction between time and intake of white chocolate (F = 0.650; p = 0.880), as shown in Fig 3.

**Table 1. Baseline demographic status of 15 healthy, pain-free women and 15 healthy, pain-free age-matched men, i.e. before injection of any of the substances.**

|  | Women (n = 15) | Men (n = 15) |
|---|---|---|
| **Age** | | |
| Mean (SD) | 25.13 (2.588) | 24.07 (3.515) |
| Min-max | 21–31 | 19–34 |
| **Stress (PSS-10)** | | |
| Median (IQR) | 3 (4) | 11 (11) |
| No stress (0–12 points) | n = 15 | n = 10 |
| Moderate degree of stress (13–20 points) | n = 0 | n = 5 |
| Severe degree of stress (21–40 pints) | n = 0 | n = 0 |
| **Depression (PHQ-9)** | | |
| Median (IQR) | 5 (4) | 3 (5) |
| Normal (0–4 points) | n = 7 | n = 10 |
| Mild (5–9 points) | n = 6 | n = 5 |
| Moderate (10–14 points) | n = 1 | n = 0 |
| Moderately severe (15–19 points) | n = 1 | n = 0 |
| Severe (20–24 points) | n = 0 | n = 0 |
| **Pain catastrophizing (PCS)** | | |
| Median (IQR) | 5 (5) | 2 (5) |
| None (0–19 pints) | n = 15 | n = 15 |
| Risk of clinical pain catastrophizing (20–29 points) | n = 0 | n = 0 |
| High risk of clinical pain catastrophizing (≥30 points) | n = 0 | n = 0 |
| **Anxiety (GAD-7)** | | |
| Median (IQR) | 4 (5) | 2 (4) |
| Normal (0–4 points) | n = 9 | n = 13 |
| Mild (5–9 points) | n = 3 | n = 2 |
| Moderate (10–14 points) | n = 1 | n = 0 |
| Severe (15–27 points) | n = 2 | n = 0 |
| **Somatization (PHQ-15)** | | |
| Median (IQR) | 18 (6) | 3 (3) |
| Normal (0–4 points) | n = 0 | n = 11 |
| Mild (5–9 points) | n = 0 | n = 4 |
| Moderate (10–14 points) | n = 1 | n = 0 |
| Severe (15–30 points) | n = 14 | n = 0 |

Data in the table are expressed as a median (interquartile range; IQR: 75th percentile minus 25th percentile). PSS-10 = Perceived Stress ScalePHQ-9 = Patient Health Questionnaire 9PCS = Pain Catastrophizing ScaleGAD-7 = Generalized Anxiety DisorderPHQ-15 = Patient Health Questionnaire 15

**Milk chocolate.** The two-way RM ANOVA showed a significant time effect (F = 103.081; $p < 0.001$), a significant difference with or without intake of milk chocolate (F = 5.065; $p = 0.032$), but no significance on the interaction between time and intake of milk chocolate (F = 0.931; $p = 0.550$). Compared to white chocolate, the experimentally induced pain intensity after intake of milk chocolate was significantly lower than without intake ($p = 0.032$), as shown in Fig 3.

In men, the two-way RM ANOVA showed a significant time effect (F = 39.806; $p < 0.001$), a significant difference with or without intake of milk chocolate (F = 4.927; $p = 0.043$), but no significance on the interaction between time and intake of milk chocolate (F = 1.059; $p = 0.392$). Compared to white chocolate, the experimentally induced pain intensity after

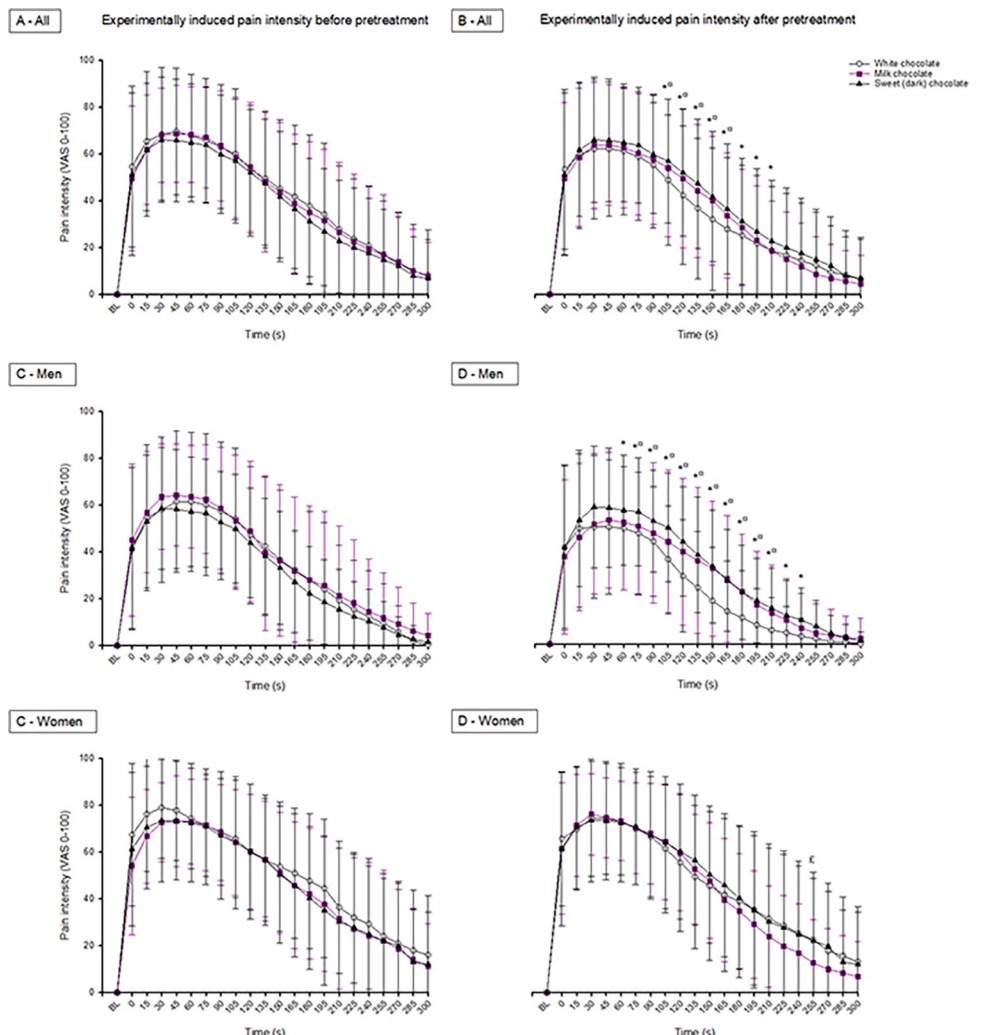

**Fig 3. Changes in pain intensity.** The mean (SEM) changes in experimentally induced pain intensity (VAS; visual analogue scale), by hypertonic saline injections, compared with baseline (BL) before and after intake of white, milk, and dark chocolate in 30 healthy, pain-free participants (A) and divided by sex into 15 women and 15 age-matched men (B-D). Assessments displayed were made every 15th s beginning immediately after injection up to 300 s after injection. The changes in pain intensity are presented both for the entire group and for the sexes separately. *Significant difference compared to baseline after intake of white chocolate (Tukey test, $p<0.05$). #Significant difference compared to baseline after intake of milk chocolate (Tukey test, $p<0.05$). §Significant difference compared to baseline after intake of dark chocolate (Tukey test, $p<0.05$).

intake of milk chocolate was significantly lower than without intake ($p = 0.044$), as shown in Fig 3.

In women, the two-way RM ANOVA showed a significant time effect (F = 66.499; $p<0.001$), but no significant difference with or without intake of milk chocolate (F = 0.863; $p = 0.369$); however, there was a significant interaction between time and intake of milk chocolate (F = 2.186; $p = 0.002$). The post-hoc test showed that the experimentally induced pain intensity after intake of milk chocolate was significantly lower than without intake at the 255 s time point after induction of pain ($p<0.05$, Tukey test), as shown in Fig 3.

**Dark chocolate.** The two-way RM ANOVA showed a significant time effect (F = 95.618; $p<0.001$), but no significant difference with or without intake of dark chocolate (F = 3.029;

$p = 0.092$); however, there was a significant interaction between time and intake of dark chocolate (F = 2.209; $p = 0.002$). The post-hoc test showed that the experimentally induced pain intensity after intake of dark chocolate was significantly lower than without intake 105–165 s after induction of pain ($p < 0.05$, Tukey test), as shown in Fig 3.

In men, the two-way RM ANOVA showed a significant time effect (F = 38.535; $p < 0.001$), a significant difference with or without intake of dark chocolate (F = 6.058; $p = 0.027$), and a significant interaction between time and intake of dark chocolate (F = 2.556; $p < 0.001$). The post-hoc test showed that the experimentally induced pain intensity after intake of dark chocolate was significantly lower than without intake 75–210 s after induction of pain ($p < 0.05$, Tukey test), as shown in Fig 3.

In women, the two-way RM ANOVA showed a significant time effect (F = 62.151; $p < 0.001$), but no significant difference with or without intake of dark chocolate (F = 0.0364; $p = 0.852$), and no significant interaction between time and intake of dark chocolate (F = 0.529; $p = 0.957$), as shown in Fig 3.

**Sex differences.** There were no sex differences at baseline in any of the assessments, i.e. before and after pre-treatment with any kind of chocolate. However, after intake of white chocolate the post-hoc test showed that men had a significantly greater reduction in pain intensity than women at 165–240 seconds after induction of pain (range of significance $p = 0.002$ to $p = 0.049$, Tukey test). There were no significant sex differences after intake of milk chocolate ($p > 0.348$, Tukey test) or dark chocolate ($p = 0.062$, Tukey test).

**Peak pain intensity.** The peak pain intensity was not significantly affected by intake of any kind of chocolate ($p > 0.05$). However, there was an overall reduction in intensity after intake of milk chocolate (6.5%) and white chocolate (3.2%), but not for dark chocolate ($p = 0.204$; Friedman RM ANOVA on ranks).

In men, intake of milk chocolate significantly reduced the peak pain intensity ($p = 0.041$; Friedman RM ANOVA on ranks); however, the reduction of peak pain intensity was not significant after white or dark chocolate intake. The intake of milk chocolate reduced the peak pain by 12.1%, while the reduction after intake of white chocolate and dark chocolate was 5.9% and 1.2%, respectively, although with no significant difference ($p > 0.05$; Tukey test).

In women, intake of any of the types of chocolate did not affect the induced peak pain intensity ($p > 0.05$). The intake of white chocolate did not affect the peak pain intensity whatsoever, while the intake of milk chocolate reduced the intensity by 1.1% and intake of dark chocolate increased peak pain intensity by 2.1% ($p = 0.786$; Friedman RM ANOVA on ranks).

When the sexes were compared, there was a significantly greater pain reducing effect after intake of milk chocolate for men than for women ($p = 0.01$), while there were no significant sex differences after intake of white or dark chocolate ($p > 0.05$).

**Pain duration.** The duration of induced pain was not significantly affected by intake of any of the chocolate types ($p > 0.05$), although it was reduced by 8.3% after intake of milk chocolate, by 14.4% after intake of white chocolate, and by 7.6% after intake of dark chocolate ($p = 0.524$; Friedman RM ANOVA on ranks).

In men, the duration of induced pain was not significantly affected by intake of any of the three chocolate types ($p > 0.05$), although duration was reduced by 5.3% after intake of milk chocolate, by 21.1% after intake of white chocolate, and by 10.0% after intake of dark chocolate ($p = 0.712$; Friedman RM ANOVA on ranks).

In women, the duration of induced pain was also not significantly affected by intake of any of the chocolate types ($p > 0.05$), although its duration was reduced by 25.0% after intake of milk chocolate and by 8.4% after intake of white chocolate, while it was unaffected after intake of dark chocolate ($p = 0.127$; Friedman RM ANOVA on ranks).

When the sexes were compared there was no significant difference in chocolate effect on pain duration after intake of any of the three chocolate types (p>0.05).

**Induced pain spread.** Intake of any of the chocolate types did not affect the induced pain spread statistically ($p>0.05$). In men, intake of any of the three chocolate types did not affect the induced pain spread ($p>0.05$). In women, intake of the chocolate types also did not affect the induced pain spread ($p>0.05$). There were no significant differences between sexes regarding chocolate effect on induced pain spread (p>0.05).

## Pressure pain threshold (PPT)

**White chocolate.** The two-way RM ANOVA showed a significant time effect (F = 11.569; $p<0.001$), a difference with or without intake of white chocolate (F = 7.344; $p = 0.011$), and an interaction between time and intake of white chocolate (F = 5.677; $p<0.001$). The post-hoc test showed that the pressure pain threshold increased significantly after intake of white chocolate when compared to no intake 15 to 30 min after induction of pain ($p<0.05$, Tukey test), as shown in Fig 4. Over the reference point, the two-way RM ANOVA showed a significant time effect (F = 3.452; $p = 0.006$), no difference with or without intake of white chocolate (F = 0.00414; $p = 0.949$), and no interaction between time and intake of white chocolate (F = 0.201; $p = 0.961$). The post-hoc test showed a significant time effect at 30 min when compared to the baseline ($p<0.05$, Tukey test).

In men, the two-way RM ANOVA showed no time effect (F = 1.716; $p = 0.142$) and no difference with or without intake of white chocolate (F = 4.518; $p = 0.052$). However, there was an interaction between time and intake of white chocolate (F = 3.272; $p = 0.010$). The post-hoc test showed that the pressure pain threshold increased significantly after intake of white chocolate when compared to no intake 5 to 30 min after induction of pain ($p<0.05$, Tukey test), as shown in Fig 4. Over the reference point, the two-way RM ANOVA showed no time effect (F = 1.750; $p = 0.135$), no difference with or without intake of white chocolate (F = 1.451; $p = 0.248$), and no interaction between time and intake of white chocolate (F = 0.658; $p = 0.657$).

In women, the two-way RM ANOVA showed a significant time effect (F = 13.399; $p<0.001$) and no difference with or without intake of white chocolate (F = 2.5708; p = 0.122). However, there was an interaction between time and intake of white chocolate (F = 4.605; $p = 0.001$). The post-hoc test showed that the pressure pain threshold increased significantly after intake of white chocolate when compared to no intake 15 to 30 min after induction of pain (p<0.05, Tukey test), as shown in Fig 4. Over the reference point, the two-way RM ANOVA showed no time effect (F = 1.765; $p = 0.131$), no difference with or without intake of white chocolate (F = 1.209; $p = 0.290$), and no interaction between time and intake of white chocolate (F = 1.072; $p = 0.383$).

**Milk chocolate.** The two-way RM ANOVA showed a significant time effect (F = 10.165; $p<0.001$), no difference with or without intake of milk chocolate (F = 4.026; $p = 0.054$), and no interaction between time and intake of milk chocolate (F = 0.779; $p<0.567$). The post-hoc test showed a significant time effect at 10 to 30 min when compared to the baseline ($p<0.05$, Tukey test), as shown in Fig 4. Over the reference point, the two-way RM ANOVA showed no time effect (F = 2.301; $p = 0.048$), no difference with or without intake of milk chocolate (F = 0.656; $p = 0.424$), and no interaction between time and intake of milk chocolate (F = 0.933; $p = 0.462$).

In men, the two-way RM ANOVA showed no time effect (F = 0.753; $p = 0.586$). However, it showed a significant difference with or without intake of milk chocolate (F = 6.250; $p = 0.025$) and a significant interaction between time and intake of milk chocolate (F = 3.297; $p = 0.010$).

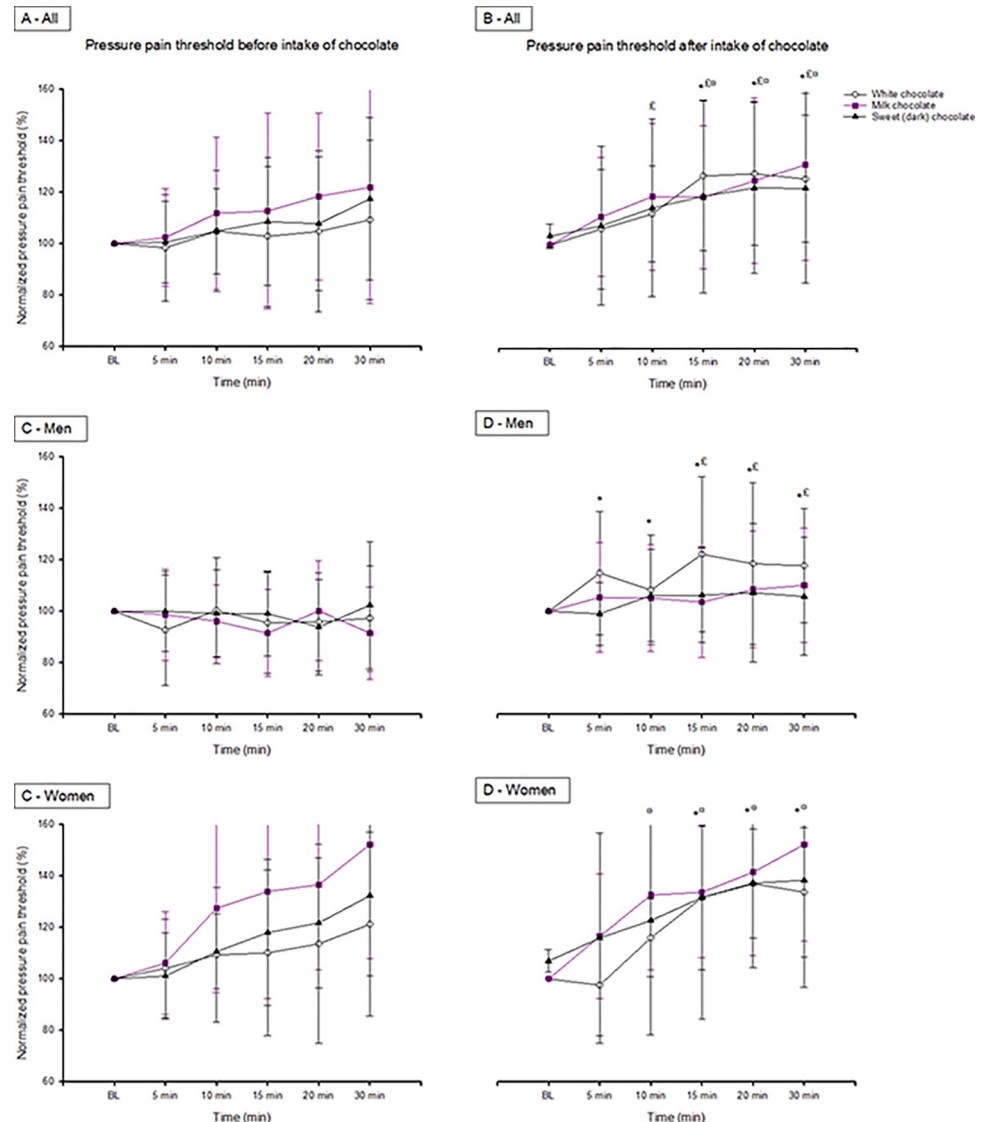

**Fig 4. Changes in pressure pain threshold.** The mean (SEM) percentage changes in pressure pain threshold (PPT; kPa), by hypertonic saline injections, compared with the baseline (BL) before and after intake of white, milk, and dark chocolate in 30 healthy, pain-free participants (A) and divided by sex into 15 women and 15 age-matched men (B-D). Assessments displayed were made every 15th s beginning immediately after injection up to 300 s after injection. The changes in pain intensity are presented both for the entire group and for the sexes separately. *Significant difference compared to baseline after intake of white chocolate (Tukey test, $p < 0.05$). #Significant difference compared to baseline after intake of milk chocolate (Tukey test, $p < 0.05$). §Significant difference compared to baseline after intake of dark chocolate (Tukey test, $p < 0.05$).

The post-hoc test showed that the pressure pain threshold increased significantly after intake of milk chocolate when compared to no intake at the 15 and 30 min time points after induction of pain ($p < 0.05$, Tukey test), as shown in Fig 4. Over the reference point, the two-way RM ANOVA showed no time effect (F = 0.806; $p = 0.549$), no difference with or without intake of milk chocolate (F = 1.282; $p = 0.277$), and no interaction between time and intake of milk chocolate (F = 1.023; $p = 0.411$).

In women, the two-way RM ANOVA showed a significant time effect (F = 19.355; $p < 0.001$), no difference with or without intake of milk chocolate (F = 0.440; $p = 0.518$), and no

interaction between time and intake of milk chocolate (F = 0.403; $p$ = 0.845), as shown in Fig 4. Over the reference point, the two-way RM ANOVA showed a significant time effect (F = 3.764; $p$ = 0.004), no difference with or without intake of milk chocolate (F = 0.379; $p$ = 0.548), and no interaction between time and intake of milk chocolate (F = 0.466; $p$ = 0.800).

**Dark chocolate.** The two-way RM ANOVA showed a significant time effect (F = 9.731; $p$<0.001), but no difference with or without intake of dark chocolate (F = 2.192; $p$ = 0.150) and no interaction between time and intake of dark chocolate (F = 1.269; $p$ = 0.280). The post-hoc test showed that the pressure pain threshold increased significantly after intake of dark chocolate when compared to no intake 15 to 30 min after induction of pain ($p$<0.05, Tukey test), as shown in Fig 4. Over the reference point, the two-way RM ANOVA showed a significant time effect (F = 3.254; $p$ = 0.008), no difference with or without intake of dark chocolate (F = 0.102; $p$ = 0.752), and no interaction between time and intake of dark chocolate (F = 0.513; $p$ = 0.766). The post-hoc test showed a significant time effect at 15 to 20 min when compared to the baseline ($p$<0.05, Tukey test).

In men, the two-way RM ANOVA showed no time effect (F = 0.568; $p$ = 0.725), no difference with or without intake of dark chocolate (F = 2.086; $p$ = 0.171), and no interaction between time and intake of dark chocolate (F = 1.247; $p$ = 0.297), as shown in Fig 4. Over the reference point, the two-way RM ANOVA showed no time effect (F = 2.351; $p$ = 0.059), no difference with or without intake of dark chocolate (F = 1.090; $p$ = 0.314), and no interaction between time and intake of dark chocolate (F = 1.025; $p$ = 0.410).

In women, the two-way RM ANOVA showed a significant time effect (F = 15.040; $p$<0.001), no difference with or without intake of dark chocolate (F = 1.090; $p$ = 0.314), and no interaction between time and intake of dark chocolate (F = 0.646; $p$ = 0.665). The post-hoc test showed that the pressure pain threshold increased significantly after intake of dark chocolate when compared to no intake 10 to 30 min after induction of pain ($p$<0.05, Tukey test), as shown in Fig 4. Over the reference point, the two-way RM ANOVA showed no time effect (F = 2.481; $p$ = 0.040), no difference with or without intake of dark chocolate (F = 0.196; $p$ = 0.665), and no interaction between time and intake of dark chocolate (F = 0.192; $p$ = 0.964).

**Sex differences.** There were no sex differences at baseline in any of the assessments, i.e. before and after pre-treatment with any type of chocolate. However, after intake of white chocolate the post-hoc test showed that men had a significantly greater increase in pressure pain threshold than women 5 min after intake, while women showed a significantly greater increase in pressure pain threshold than men 20 min after intake ($p$<0.05, Tukey test). When it comes to milk chocolate, women showed a significantly greater increase in pressure pain threshold than men 10 to 30 min after intake ($p$<0.05, Tukey test). Also, after intake of dark chocolate women showed a greater increase in pressure pain threshold than men 15 to 30 min after intake ($p$<0.05, Tukey test). Finally, there were no sex differences at any time point over the reference point regardless of chocolate type.

## Discussion

The main finding was that all administered types of chocolate, i.e., white (30% cocoa), milk (34% cocoa) and dark (70% cocoa) resulted in a reduction of the experimentally induced pain using the same experimental procedures as described in other studies [11, 48–50]. However, the cocoa content does not seem to be the main reason for reducing pain, disproving our hypothesis that the higher the cocoa content, the less pain experienced. Rather, the results indicate that it might be other ingredients in the chocolate such as sugar that explain these chocolate-induced reduction of pain variables.

In addition, PPT does not seem to be affected by intake of any type of chocolate, although this study could show some significant differences in PPT after intake of chocolate among either men or women. However, this is in line with previous studies which also have shown that PPT is not affected by various pain models and treatments [13, 51].

## Function of cocoa

Previous studies have suggested chocolate as a complement to the traditional medical treatment of pain by inhibiting induced neurogenic inflammatory responses. However, these studies examined the effect of chocolate by using a neurogenic pain model on rats and not in intramuscularly induced pain in humans [36, 37]. In this present study, the pain reducing effect could not be fully attributed to the anti-inflammatory effects of cocoa, as found in a study showing that certain flavanols regulate the anti-inflammatory cytokine levels of IL-4 and TGF-β [36].

Cocoa-derived products that are rich in flavanols have been shown to reduce inflammation [36]. More specifically, flavonoids (type of antioxidant), which are found in certain fruits, vegetables, and in high concentrations in cocoa have been studied [34, 36, 52]. Also, as mentioned, intake of the amino acid tryptophan plays an important role in serotonin synthesis. However, daily intake of tryptophan needs to be at least six grams to have an increased effect on serotonin synthesis and subsequent mood effects [25], while this study used 3.6 g. Therefore, the amount of cocoa consumed during the experiments might be of importance to the outcome of the result. The small variations in pain experienced when comparing the different types of chocolate in this present study may have been due to the limited amount of chocolate that was consumed [36].

## Differences between the chocolates and their sugar content

The sugar content in the different types of chocolate, mostly in the form of sucrose, has been shown to have an increased analgesic effect, and white chocolate tends to have a higher percentage of sugar than milk or dark chocolate. The increased effect might be due to the mechanism of release of opioids [53], with chocolate having a potentially similar mechanism of neurological addiction as other substances [54]. Given the potential effect of sugar content in chocolate on pain, both white and milk chocolate had a greater effect on reducing pain in the current study than dark chocolate which has a higher cocoa content. Milk chocolate also had a pain reducing effect regarding peak pain intensity and pain area.

This pain reducing effect is also in accordance with other studies [53, 55–57]. In these studies, the administration of sugar (e.g., sucrose, a disaccharide composed of glucose and fructose) was followed by a cold pressure test and the results showed an effect on pain tolerance and pain sensitivity [55]. A positive experience regarding the taste of different foods can stimulate mechanisms in the brain to release endorphins and neurotransmitters that result in increased pain tolerance [53]. The results of the chocolate ratings suggested the same, since white chocolate and milk chocolate had almost the same amount of "3" ratings, meaning they were preferred by the participants and both have higher amount of sugar than dark chocolate. And milk chocolate was the most pleasing according to the participants' preferences, which might be one possible explanation as to why milk chocolate samples had a more significant effect on the pain intensity (but not duration) compared to white chocolate, even though these two consisted of a similar amount of cocoa.

But other studies found opposing results [36, 58]. However, these studies utilized a cocoa-enriched diet over a longer period of time, and one of them used a dose equivalent to a daily consumption of 33 g of cocoa powder for 14 days [58], which could also potentially affect the

outcome compared to this study, which administered 3.6 g of chocolate per participant. The long-term ingestion of cocoa high in flavanols causes a change in platelet function similar to that of aspirin, becoming less potent [36, 59, 60], although this was not tested in the present study.

The fact that dark chocolate got less significant results may be because of the bitter taste, and less sugar content [53, 56]. Studies have reported significant differences in pain tolerance with sucrose water compared to cocoa infused solution, which showed no increase in pain tolerance [53]. This contrasts with the expectation that cocoa can be used for pain reducing effects [35, 36, 53].

Furthermore, in studies investigating the effect of chocolate on cold pain tolerance it was shown that preference, not cocoa concentration, was responsible for changes in tolerance [53]. Thus, based on the findings from the present study, and other studies [35, 36, 53], it would be intriguing to evaluate if a chocolate with a different balance of sweetness and cocoa concentration can affect pain tolerance more significantly than just sweet foods without any cocoa.

## Study strengths and limitations

A significant strength of this study is that it was designed as a randomized double-blinded study, with both male and female participants within a small age-range. All the participants in the study fully completed the questionnaires and experiments, with no missing data.

Pain induction by hypertonic saline has been widely used in several studies due to its ability to mimic clinical acute muscle pain [11]. In this study, the induced saline injection caused a deep masseter muscle pain (63-70/100) of moderate intensity that spread to various other regions, such as the teeth; this is also shown by other studies using experimentally induced pain as a pain model [11, 51, 53]. Hence, another strength is that the induced pain in the present study had an intensity that can be considered as clinically relevant.

Despite its findings, this study is not without limitations. We did not control for any ingestion of sweet foods before the visit, which could have affected the results [53]. However, probable food intake by some participants before the visit should not have affected the general results, since participants act as their own control at each visit and the results were so similar between participants. Another limitation is the time after ingestion of chocolate. Perhaps the time after ingestion was not enough to achieve any flavanol modulation of neurogenic inflammation. We also did not control for the composition of the chocolate types, such as soy, or vanilla content, which indeed could have influenced the outcome, as results indicate that it might be dependent on the individual's preference and taste-experience, or even other ingredients in the chocolate that could explain this chocolate-induced reduction of pain variables.

One important confounding factor is that once the participant has been given the chocolate and taste it, they likely know what type of chocolate they were eating. Their preference could interfere with results. On the other hand, in this study preference and taste-experience were included in the results, which might have helped to address the limitations mentioned above.

As mentioned before, another limitation was the amount of chocolate that was given to the participants (3.6 g), which is much lower than other studies exploring the effect of flavanol or tryptophan content [36, 40]. Lastly, the data presented here could have been analyzed using a full-scale crossover design, with a mixed-effect model with a period effect, and using other software like SAS (SAS Institute, Cary NC). Although we took advantage of a cross-over design by utilizing each subject as their own control and a smaller number of patients, the duration of the study was likely short compared to a full-scale design, and the washout period might not have been long enough to remove the effect from the different types of chocolate on one another. We did base our analytical choice on our previous studies using the same

experimental methodology and also on several others using SPSS for the same study design [61]. Yet, we cannot claim that the chosen design and statistical analyses were infallible.

## Future studies

Future studies using a cross-over design should investigate the significance of cocoa as a factor in pain experience, including the amount ingested and the duration of analgesic effects after ingestion. For example, chocolates similar in taste and sugar/sweetness content, but with different cocoa concentration (70% vs 30–34%), should be used to investigate the extent to which the cocoa itself influences the perceived pain. Lastly, studies should compare commercially available chocolate with the ones used in this study, as they differ in cocoa content, particularly the white chocolate and to some extent the milk chocolate.

## Conclusion

This study showed that intake of any type of chocolate 5 minutes before a painful stimulus has a pain reducing effect no matter the cocoa concentration. The results indicate that perhaps it is not the cocoa concentration (e.g., flavanols) alone that explains the positive effect on pain, but likely a combination of preference and taste-experience. Another possible explanation could be the composition of the chocolate, i.e. the concentration of the other ingredients such as sugar, soy, and vanilla.

## Supporting information

**S1 Checklist.**
(DOC)

**S1 Data.**
(XLSX)

**S1 File.**
(DOCX)

## Author Contributions

**Conceptualization:** Mario Brondani, Essam Ahmed Al-Moraissi, Sofia Louca Jounger, Nikolaos Christidis.

**Data curation:** Alexandra Hajati, Bruna Brondani, Nikolaos Christidis.

**Formal analysis:** Alexandra Hajati, Lina Angerstig, Victoria Klein, Linda Liljeblad, Essam Ahmed Al-Moraissi, Bruna Brondani, Nikolaos Christidis.

**Investigation:** Alexandra Hajati, Lina Angerstig, Victoria Klein, Linda Liljeblad.

**Methodology:** Mario Brondani, Sofia Louca Jounger, Nikolaos Christidis.

**Resources:** Nikolaos Christidis.

**Supervision:** Mario Brondani, Sofia Louca Jounger, Nikolaos Christidis.

**Writing – original draft:** Alexandra Hajati.

**Writing – review & editing:** Mario Brondani, Lina Angerstig, Victoria Klein, Linda Liljeblad, Essam Ahmed Al-Moraissi, Sofia Louca Jounger, Bruna Brondani, Nikolaos Christidis.

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
