## [Decision Letter · Decision Letter 0]

25 Nov 2022

PONE-D-22-13220Cocoa intake and muscle pain sensation: a randomized experimental studyPLOS ONE

Dear Dr. Christidis,

Thank you for submitting your manuscript to PLOS ONE. After careful consideration, we feel that it has merit but does not fully meet PLOS ONE’s publication criteria as it currently stands. Therefore, we invite you to submit a revised version of the manuscript that addresses the points raised during the review process.

The reviewerrìs raised serious critiques to some key methodological aspects and also on other parts of the manuscript. Please assess if you are able to address their concerns appropriately. 

We look forward to receiving your revised manuscript.

Kind regards,

Andrea Martinuzzi

Academic Editor

PLOS ONE

Journal Requirements:

2. Thank you for submitting your clinical trial to PLOS ONE and for providing the name of the registry and the registration number. The information in the registry entry suggests that your trial was registered after patient recruitment began. PLOS ONE strongly encourages authors to register all trials before recruiting the first participant in a study.

a) your reasons for your delay in registering this study (after enrolment of participants started);

b) confirmation that all related trials are registered by stating: “The authors confirm that all ongoing and related trials for this drug/intervention are registered”.

a) If there are ethical or legal restrictions on sharing a de-identified data set, please explain them in detail (e.g., data contain potentially identifying or sensitive patient information) and who has imposed them (e.g., an ethics committee). Please also provide contact information for a data access committee, ethics committee, or other institutional body to which data requests may be sent. Note that it is not acceptable for an author to be the sole named individual responsible for ensuring data access.

Reviewers' comments:

Reviewer's Responses to Questions

**Comments to the Author**

1. Is the manuscript technically sound, and do the data support the conclusions?

Reviewer #1: Partly

Reviewer #2: Partly

Reviewer #3: No

2. Has the statistical analysis been performed appropriately and rigorously? 

Reviewer #1: No

Reviewer #2: Yes

Reviewer #3: I Don't Know

3. Have the authors made all data underlying the findings in their manuscript fully available?

Reviewer #1: No

Reviewer #2: Yes

Reviewer #3: Yes

4. Is the manuscript presented in an intelligible fashion and written in standard English?

Reviewer #1: Yes

Reviewer #2: No

Reviewer #3: Yes

5. Review Comments to the Author

Reviewer #1: *** General comments: ***

The design is a three-period three-treatment crossover design with the further complication of including a two-factor within-subject design with multiple evaluations over time in each period. This design requires special techniques to analyze correctly. See Senn (2002) for details about methodology. The current analysis is incorrect.

The manuscript lacks line numbers, therefore, it is difficult to provide specific suggestions. There are minor grammatic errors that might benefit from a third-party proof-reading.

In addition, it does not seem that the data are available for review.

*** Specific comments: ***

The authors presumably have already found that their design is not going to be handled correctly using SigmaPlot -- however, the repeated measures analysis that is proposed is incorrect. Also, it seems likely that the implied R-side correlation structure implemented by SigmaPlot is too simplistic. Further, the use of the Friedman test is even more limited in its ability to correctly analyze these data.

This crossover design features:

* 3 periods

* 3 treatments

* 6 sequences

* 30 subjects

The within period design features:

* 2 conditions (pre- and post-chocolate)

* Multiple time-points

* Multiple measures

While it is not clear that performing a correct analysis would lead to substantially different results, it is difficult to say ahead of actually performing it. A correct analysis will include treatment, sequence, subject, period, and carryover effects in the between-subject design, and pre/post and time in the within-subject design. It may be reasonable to omit the carryover, but this must be justified. Senn (2002) provides an excellent discussion of this issue that may be of help.

As mentioned in the limitations, there is most likely a period effect. Therefore, this must be accounted in the analysis at a minimum. The randomization restriction represented by the sequence is also important. A subject-level effect is required to produce the correct degrees of freedom for analysis (and potentially, depending on how the analysis is specified, a random effect of subject by period). Most likely, the authors would like to generalize their results past this specific set of subjects, so that subject should be considered as a random effect.

This design should probably be analyzed using either SAS or R. It is recommended to use SAS, unless the authors have access to expertise in using R to specify complex analyses. In SAS it would be feasible to implement the entire design as run using PROC MIXED. However, setting up the analysis does require a little experience, effort, and likely some trial and error and it would require both G-side and R-side specifications.

Having set this analysis up would provide several benefits. First, it allows all tests of interests to be performed as either omnibus tests or contrasts (whether corrected for multiple testing or not). Second, it allows covariates to be entered to evaluate their effects net of the treatment or vice versa. Third, it provides a template that correctly reflects the experiment design in all its complexity and can be used for all assessments (ideally).

Alternatively, while the fine-grained sampling over time is interesting, it seems to provide little benefit in the actual analysis. Therefore, a simpler method of analyzing the data would be as follows:

* Summarize the data across time using a statistic or statistics, e.g.:

- Area under the curve

- Peak value

- Other

* Determine the differential effect of chocolate using the summary statistic, e.g.:

- Post-chocolate minus pre-chocolate

- Ratio of post-chocolate to pre-chocolate

* Perform a standard 3-treatment 3-period crossover analysis

Some sample PROC MIXED code for this "simpler" analysis would be:

PROC MIXED;

CLASS SEQUENCE SUBJECT PERIOD TREATMENT;

MODEL DELTA = TREATMENT PERIOD;

RANDOM SUBJECT(SEQUENCE);

LSMEANS TREATMENT / PDIFF CL E;

RUN;

(Note that the code above could be extended to allow the pre/post x time design within subjects, and this would indeed be the starting point for that effort.)

A side benefit of this approach is that it allows an easier comparison of the various measures with each other via scatterplots or correlation analysis. It is also more likely to provide data that meet the assumptions of the linear model analysis.

The authors state that they evaluate normality of the data. This is a misnomer. One can easily see that in any case of interest with actual group differences that the data will not be normally distributed. What is assessed is the normality of the residuals from the linear model analysis. (Or, as in this case, suitably standardized or studentized residuals.)

Even if these residuals are not completely consistent with normality, it is usually preferable to use the same analysis structure throughout rather than a hodge-podge of statistical methods. Nonparametric analogues can be used as support for this. Recall that linear models still produce unbiased results even if error distributions are not normal. This pays dividends in interpretability and ease of representing the experimental design.

Note that the authors present significance tests based on ranks, but then quantitative results based on means. This is a common method to try to dodge the fact that rank-based methods provide no real quantitative statements but strikes one as disingenuous. Using a linear model approach is better for this purpose, even if the model fit is only fair.

*** Reference: ***

Senn, S. S. (2002). Cross-over trials in clinical research (Vol. 5). John Wiley & Sons.

Reviewer #2: This paper reports on a generally well-designed study producing novel data that intake of chocolate reduces acute pain induced by hypertonic saline injection into the masseter muscle of young healthy men and women. There are however several points that need to be satisfactorily addressed by the authors before it could be considered acceptable.

1. The first 4 paragraphs of the Introduction inexplicably focus on chronic pain, yet this study did not address chronic pain per se since it used an experimental model of acute pain which the Discussion states is used “to mimic clinical acute muscle pain”. The 1st part of the Introduction thus needs to be re-cast with a focus on acute pain and underlying mechanisms, including those expressed in the masticatory muscles.

2. The introduction’ s middle paragraph about cocoa inhibiting “trigeminal nerve cells……spinal cord” needs modification since trigeminal nerve cells predominantly project to the brainstem, not spinal cord.

3. There seems to be a disconnect in the description of the Study Design. Clearer wording is needed. Items in pain scores are said to have been graded on a 0-10 scale but this is followed by the statement that they were graded from 0 to 4.

4. Timelines are given in the Methods text and Flow Chart but It is not convincing that there was sufficient time for chocolate ingestion to have occurred to exert analgesic effects (eg, via modulation of neurogenic inflammation) especially during the initial time points where the pain reductions are extremely modest. Also the time designation on the Time axis in Figs 3 and 4 are tiny and very difficult for the reader with even 20/20 vision to discern.

5. The Study Design outlines measures to assess both pain intensity and pain disability, but the Results overlooks any presentation of the latter.

6. Strengths of the study included the use of different concentration of cocoa in the chocolate pieces ingested, and the finding that taste preference played a key role. The 1st couple of paragraphs of the Discussion discuss cocoa and its analgesic effects but it is unclear how a definitive conclusion could be reached from the present study that it is the cocoa in the ingested pieces of chocolate that was responsible for the modest analgesic effects. There were other ingredients in the chocolate (soy, sugar, vanilla) and there is no indication in the study design that these were controlled for, despite different cocoa concentration being used. Accordingly, the title of the paper should be modified to reflect “chocolate” not ‘cocoa’.

7. The !st sentence of the Discussion does not make sense; it seems to be missing words.

8. Likewise, the 1st sentence of the 9th paragraph about taste preference being “responsible for the response to cold pain tolerance” is unclear and needs to be re-formulated to make it clear how taste preference could affect a “response” to the “tolerance” of “cold pain”. The next 2 sentences also have a problem in the formatting (and show grammatical errors).

9. The Study Limitations will need to include those identified above (eg, Point 4).

Reviewer #3: The subject could be of great interest however, the design and the interpretation of the results are a limitation. The main concern is that the authors do not know or not show the composition of the chocolate. White chocolate has only cocoa fat with no flavanols at all (30% of cocoa fat is a lot really). They omit the sugar content in the different chocolates that should be around 50% for milk and white chocolate. Maybe these facts can better explain the preference and the un-expected results regarding the "higher flavanols the higher analgesic effect". I think that, unfortunately, it is not possible to get to any conclusion because there is no real control for the effect. It looks to me that it is the sugar content the one producing any effect and to probe it you would need a non -sugar content control

When authors say that is a controlled study: Which is the control?

The conclusion “The results indicate that it is not mainly the cocoa-concentration that has a positive effect on pain-characteristics, but the effect seems to be affected by the preference” is not based in the results. There is no significance or correlation to support it. Again I think the difference lays in the sugar content.

I don’t see the point of using an abbreviation for serotonin.

Did the authors check for a sequence effect in the assays (white, milk or dark) on the preference for one or other chocolate?

6. PLOS authors have the option to publish the peer review history of their article (what does this mean?). If published, this will include your full peer review and any attached files.

Reviewer #1: No

Reviewer #2: No

Reviewer #3: No

---

## [Author Response · Author response to Decision Letter 0]

13 Dec 2022

Response to reviewers:

PONE-D-22-13220

Cocoa intake and muscle pain sensation: a randomized experimental study

Nikolaos Christidis, 2022-12-04

Thank you for considering this manuscript for publication. Since two of the reviewers approved the statistics and the third had some concerns, we have tried to address some parts of the statistics. The methods and set-up for this study have been used in several (hundreds) publications with hypertonic saline as experimental pain instigator, so any carry over effects have been taken into consideration in the study plan, thus not affecting the results. Furthermore, we have several publications regarding this model using the same statistics after consultation with a biomedical statistician, and this statistical method has also been approved in three doctoral theses from our group alone (Christidis Nikolaos 2010, Sofia Louca Jounger 2017, and Samaa Al Sayegh 2021). However, as it was suggested to use one statistical method for all analyses, we discussed using 2-way repeated measures for all analyses. Since data was not normally distributed we chose the more conservative non-parametric statistical Friedman repeated measures ANOVA method. In fact, we have already presented the pre-and post-chocolate effect difference for the analysis; we have also used the peak pain value for analysis, but since it was not obvious for the reviewer we have clarified that section. Finally, we do not have the statistical programs suggested by the reviewer, so we have to use SigmaPlot. Therefore, we hope that you still consider this manuscript based on the fact that the method used has been approved several times before. 

We would like to thank all three reviewers for the comments and suggested changes in order to improve the quality of this manuscript. The changes are highlighted in yellow. We hope that this reply and changes in the manuscript are sufficient and will answer your queries. 

Reviewer 1: 

*** General comments: ***

The design is a three-period three-treatment crossover design with the further complication of including a two-factor within-subject design with multiple evaluations over time in each period. This design requires special techniques to analyze correctly. See Senn (2002) for details about methodology. The current analysis is incorrect.

The manuscript lacks line numbers, therefore, it is difficult to provide specific suggestions. There are minor grammatic errors that might benefit from a third-party proof-reading.

In addition, it does not seem that the data are available for review.

*** Specific comments: ***

The authors presumably have already found that their design is not going to be handled correctly using SigmaPlot -- however, the repeated measures analysis that is proposed is incorrect. Also, it seems likely that the implied R-side correlation structure implemented by SigmaPlot is too simplistic. Further, the use of the Friedman test is even more limited in its ability to correctly analyze these data.

This crossover design features:

* 3 periods

* 3 treatments

* 6 sequences

* 30 subjects

The within period design features:

* 2 conditions (pre- and post-chocolate)

* Multiple time-points

* Multiple measures

While it is not clear that performing a correct analysis would lead to substantially different results, it is difficult to say ahead of actually performing it. A correct analysis will include treatment, sequence, subject, period, and carryover effects in the between-subject design, and pre/post and time in the within-subject design. It may be reasonable to omit the carryover, but this must be justified. Senn (2002) provides an excellent discussion of this issue that may be of help.

As mentioned in the limitations, there is most likely a period effect. Therefore, this must be accounted in the analysis at a minimum. The randomization restriction represented by the sequence is also important. A subject-level effect is required to produce the correct degrees of freedom for analysis (and potentially, depending on how the analysis is specified, a random effect of subject by period). Most likely, the authors would like to generalize their results past this specific set of subjects, so that subject should be considered as a random effect.

This design should probably be analyzed using either SAS or R. It is recommended to use SAS, unless the authors have access to expertise in using R to specify complex analyses. In SAS it would be feasible to implement the entire design as run using PROC MIXED. However, setting up the analysis does require a little experience, effort, and likely some trial and error and it would require both G-side and R-side specifications.

Having set this analysis up would provide several benefits. First, it allows all tests of interests to be performed as either omnibus tests or contrasts (whether corrected for multiple testing or not). Second, it allows covariates to be entered to evaluate their effects net of the treatment or vice versa. Third, it provides a template that correctly reflects the experiment design in all its complexity and can be used for all assessments (ideally).

Alternatively, while the fine-grained sampling over time is interesting, it seems to provide little benefit in the actual analysis. Therefore, a simpler method of analyzing the data would be as follows:

* Summarize the data across time using a statistic or statistics, e.g.:

- Area under the curve

- Peak value

- Other

* Determine the differential effect of chocolate using the summary statistic, e.g.:

- Post-chocolate minus pre-chocolate

- Ratio of post-chocolate to pre-chocolate

* Perform a standard 3-treatment 3-period crossover analysis

Some sample PROC MIXED code for this "simpler" analysis would be:

PROC MIXED;

CLASS SEQUENCE SUBJECT PERIOD TREATMENT;

MODEL DELTA = TREATMENT PERIOD;

RANDOM SUBJECT(SEQUENCE);

LSMEANS TREATMENT / PDIFF CL E;

RUN;

(Note that the code above could be extended to allow the pre/post x time design within subjects, and this would indeed be the starting point for that effort.)

A side benefit of this approach is that it allows an easier comparison of the various measures with each other via scatterplots or correlation analysis. It is also more likely to provide data that meet the assumptions of the linear model analysis.

The authors state that they evaluate normality of the data. This is a misnomer. One can easily see that in any case of interest with actual group differences that the data will not be normally distributed. What is assessed is the normality of the residuals from the linear model analysis. (Or, as in this case, suitably standardized or studentized residuals.)

Even if these residuals are not completely consistent with normality, it is usually preferable to use the same analysis structure throughout rather than a hodge-podge of statistical methods. Nonparametric analogues can be used as support for this. Recall that linear models still produce unbiased results even if error distributions are not normal. This pays dividends in interpretability and ease of representing the experimental design.

Note that the authors present significance tests based on ranks, but then quantitative results based on means. This is a common method to try to dodge the fact that rank-based methods provide no real quantitative statements but strikes one as disingenuous. Using a linear model approach is better for this purpose, even if the model fit is only fair.

*** Reference: ***

Senn, S. S. (2002). Cross-over trials in clinical research (Vol. 5). John Wiley & Sons. 

Reply: 

Thank you for reviewing this manuscript. We have added line-numbers in the version with the changes tracked. 

When it comes to the concerns it seems that your major concern is the statistical method used. We have also been told by our biostatisticians to use non-parametric statistical methods for not normally distributed data as well as categorical and ranked data, even though they are more conservative. That is why we have used the Friedman repeated measures test for peak pain, pain duration and pain area. 

Some of your concerns regard a carry-over effects. In this case each subject is its own control so we have used the pre- and post-chocolate difference as you suggested. However, since this was not obvious, we have clarified it in the intake of chocolate session as well as in the statistics section. 

You also suggested that we should use simpler statistics such as the peak pain intensity difference, which already is presented in the results under its own section.

Further, there are thousands of publications on hypertonic saline as experimental setting and our group has used this method in several studies and in three theses that have been approved. 

A final reason for not using more tests is that we do not wish to “significance-fish”.

Unfortunately, we do not have the statistical programs suggested, nor the knowledge to use the R system. Therefore, we use the analysis previously used and approved as suggested by our statisticians. We hope this response, where we consider your suggestion to use one method instead of mixing, where we have the pre- and post-chocolate difference, and the clarification in the text is acceptable for you.

Reviewer 2:

This paper reports on a generally well-designed study producing novel data that intake of chocolate reduces acute pain induced by hypertonic saline injection into the masseter muscle of young healthy men and women. There are however several points that need to be satisfactorily addressed by the authors before it could be considered acceptable.

1. The first 4 paragraphs of the Introduction inexplicably focus on chronic pain, yet this study did not address chronic pain per se since it used an experimental model of acute pain which the Discussion states is used “to mimic clinical acute muscle pain”. The 1st part of the Introduction thus needs to be re-cast with a focus on acute pain and underlying mechanisms, including those expressed in the masticatory muscles.

Reply: Thank you for this comment. The first part of the introduction has changed focus to acute, experimental pain as suggested.

2. The introduction’ s middle paragraph about cocoa inhibiting “trigeminal nerve cells……spinal cord” needs modification since trigeminal nerve cells predominantly project to the brainstem, not spinal cord.

Reply: This has been corrected accordingly.

3. There seems to be a disconnect in the description of the Study Design. Clearer wording is needed. Items in pain scores are said to have been graded on a 0-10 scale but this is followed by the statement that they were graded from 0 to 4.

Reply: This part has been re-written to clarify any concerns.

4. Timelines are given in the Methods text and Flow Chart but It is not convincing that there was sufficient time for chocolate ingestion to have occurred to exert analgesic effects (eg, via modulation of neurogenic inflammation) especially during the initial time points where the pain reductions are extremely modest. Also, the time designation on the Time axis in Figs 3 and 4 are tiny and very difficult for the reader with even 20/20 vision to discern.

Reply: You are right; however, since this was the first study of its kind we did not know and chose to use the standard time-points for this experimental pain model. Perhaps time was not sufficient, but that requires a new study with different timelines. This is addressed in study limitations.

The figures might have been downsized in the system for the review. They have a high resolution of 900 dpi and can be zoomed in easily.

5. The Study Design outlines measures to assess both pain intensity and pain disability, but the Results overlooks any presentation of the latter.

Reply: This was misleading in the study design; no such assessment or analysis was performed since the study aimed at pain characteristics. This has been clarified. 

6. Strengths of the study included the use of different concentration of cocoa in the chocolate pieces ingested, and the finding that taste preference played a key role. The 1st couple of paragraphs of the Discussion discuss cocoa and its analgesic effects but it is unclear how a definitive conclusion could be reached from the present study that it is the cocoa in the ingested pieces of chocolate that was responsible for the modest analgesic effects. There were other ingredients in the chocolate (soy, sugar, vanilla) and there is no indication in the study design that these were controlled for, despite different cocoa concentration being used. Accordingly, the title of the paper should be modified to reflect “chocolate” not ‘cocoa’.

Reply: The title has been changed accordingly

7. The 1st sentence of the Discussion does not make sense; it seems to be missing words.

Reply: This has been adjusted accordingly

8. Likewise, the 1st sentence of the 9th paragraph about taste preference being “responsible for the response to cold pain tolerance” is unclear and needs to be re-formulated to make it clear how taste preference could affect a “response” to the “tolerance” of “cold pain”. The next 2 sentences also have a problem in the formatting (and show grammatical errors).

Reply: This part has been adjusted as suggested. 

9. The Study Limitations will need to include those identified above (eg, Point 4).

Reply: This has been added as you pointed out.

Reviewer 3:

The subject could be of great interest however, the design and the interpretation of the results are a limitation. 

The main concern is that the authors do not know or not show the composition of the chocolate. White chocolate has only cocoa fat with no flavanols at all (30% of cocoa fat is a lot really). They omit the sugar content in the different chocolates that should be around 50% for milk and white chocolate. Maybe these facts can better explain the preference and the un-expected results regarding the "higher flavanols the higher analgesic effect". I think that, unfortunately, it is not possible to get to any conclusion because there is no real control for the effect. It looks to me that it is the sugar content the one producing any effect and to probe it you would need a non -sugar content control

Reply: We have added the composition of the chocolates and clarified in the first paragraph of the discussion that there are other, more probable factors that can explain the positive outcome, as suggested. This was also addressed in the study limitations and conclusion (as suggested further down).

When authors say that is a controlled study: Which is the control?

Reply: Each participant is its own control. First, they get an injection with hypertonic saline, then after half an hour chocolate, and 2 minutes later a new injection with hypertonic saline. Therefore, the control is pre-chocolate injection vs post-chocolate injection in each subject and the analysis is based on that. This has been clarified according to your question.

The conclusion “The results indicate that it is not mainly the cocoa-concentration that has a positive effect on pain-characteristics, but the effect seems to be affected by the preference” is not based in the results. There is no significance or correlation to support it. Again I think the difference lays in the sugar content.

Reply: This has been adjusted as mentioned above.

I don’t see the point of using an abbreviation for serotonin.

Reply: This has been changed, as suggested.

Did the authors check for a sequence effect in the assays (white, milk or dark) on the preference for one or other chocolate?

Reply: Unfortunately, that data does not exist for one of the sexes, just which one they preferred. That is why we presented it in the “Demographic data”.

---

## [Decision Letter · Decision Letter 1]

3 Feb 2023

PONE-D-22-13220R1Chocolate intake and muscle pain sensation: a randomized experimental studyPLOS ONE

Dear Dr. Christidis,

Thank you for submitting your manuscript to PLOS ONE. After careful consideration, we feel that it has merit but does not fully meet PLOS ONE’s publication criteria as it currently stands. Therefore, we invite you to submit a revised version of the manuscript that addresses the points raised during the review process.

The reviewers still highllight some major problems in the statistical analysis that critically undermine the validity of the work. Consider if you are able to address the issue otherwise consider withdrawal and possibly resumbission under different headline or journal.

We look forward to receiving your revised manuscript.

Kind regards,

Andrea Martinuzzi

Academic Editor

PLOS ONE

Reviewers' comments:

Reviewer's Responses to Questions

**Comments to the Author**

1. If the authors have adequately addressed your comments raised in a previous round of review and you feel that this manuscript is now acceptable for publication, you may indicate that here to bypass the “Comments to the Author” section, enter your conflict of interest statement in the “Confidential to Editor” section, and submit your "Accept" recommendation.

Reviewer #2: (No Response)

Reviewer #3: (No Response)

Reviewer #4: (No Response)

2. Is the manuscript technically sound, and do the data support the conclusions?

Reviewer #2: Yes

Reviewer #3: Partly

Reviewer #4: Partly

3. Has the statistical analysis been performed appropriately and rigorously? 

Reviewer #2: Yes

Reviewer #3: I Don't Know

Reviewer #4: No

4. Have the authors made all data underlying the findings in their manuscript fully available?

Reviewer #2: Yes

Reviewer #3: No

Reviewer #4: No

5. Is the manuscript presented in an intelligible fashion and written in standard English?

Reviewer #2: Yes

Reviewer #3: Yes

Reviewer #4: Yes

6. Review Comments to the Author

Reviewer #2: The authors have satisfactorily addressed my remaining points from the previous review, except for point 7 since the first sentence still is unclear: how could it be that findings that “the pain reduction by the different types of chocolate presented in this study is line with previous studies “ (11, 48-50); none of these 4 cited studies investigated the effects of chocolate on pain. The sentence seems unnecessary and could be deleted unless the authors modify this sentence to clarify exactly what they mean by this unclear sentence.

Also note the revised wording on line 211 has introduced inappropriate tense changes in the same sentence; past tense should be used for both sets of verbs.

Reviewer #3: I think that the composition of the chocolates, now that it has been included, clearly shows that the effect, if any, is not due to the bioactives in cocoa, polyphenols and methylxanthines, but to maybe the cocoa fat and the sugar. In fact, the results show that in the case of men the sugar and fat content with no cocoa bioactives is more effective than the bioactive containing cocoa of dark chocolate.

I think that all this should be included in the abstract. It does not seem to be a question of preference but a question of sugar (and maybe fat contents)

This is in disagreement with most papers showing an effect of cocoa polyphenols and methylxanthines at different levels

See for instance:

Goya L, Kongor JE, de Pascual-Teresa S. From Cocoa to Chocolate: Effect of Processing on Flavanols and Methylxanthines and Their Mechanisms of Action. Int J Mol Sci. 2022;23(22):14365. Published 2022 Nov 18. doi:10.3390/ijms23221436

or

Gu Y, Lambert JD. Modulation of metabolic syndrome-related inflammation by cocoa. Mol Nutr Food Res. 2013;57(6):948-961. doi:10.1002/mnfr.201200837

etc

However, I think the paper should be published showing the results they got with really low amounts of cocoa solids (0% in white chocolate and only 9% in milk chocolate).

It is very important to show that this assay was done based on the bioactivity of cocoa (polyphenols and methylxanthines) however, the real content of this compounds, that was not determined, should be really low since cocoa fat has not beneficial effects that we knew until now.

Reviewer #4: The paper presents a randomized double-blinded controlled trial. First and foremost, the most important question if this is a powered trial to detect a significant effect. With 30 samples and without any power analysis this is not a powered trial. So any claim of significance based on p-value does not make sense.

Second, I reviewed the previous round of reviews and agree with the statistical reviewer that this should be analyzed using a crossover design. A mixed-effect model with a period effect should be used. The previous reviewer has given some excellent guidelines on how to analyze the data which I agree with. He/she also has SAS code, which was pretty tough. Unfortunately, all the suggestions are ignored for two reasoning. 1. They do not have access to SAS and 2. The authors have used similar simpler methods earlier which are accepted.

This is not a constructive way to address the reviewer’s suggestions. First, if SAS is not available, please use R, which is completely free to use. Second, just because previous papers are not reviewed by a statistician thoroughly it does not mean the methods used by them are all appropriate. It seems the investigative teams lack a statistician as otherwise converting the given SAS code to R and reanalyzing the data using a crossover design is not a big issue. The reviewer all mentions several other challenges that have to do with normality and model fitting which I cannot comment on given the suggestion is ignored.

7. PLOS authors have the option to publish the peer review history of their article (what does this mean?). If published, this will include your full peer review and any attached files.

Reviewer #2: No

Reviewer #3: No

Reviewer #4: No

---

## [Author Response · Author response to Decision Letter 1]

17 Feb 2023

Response to reviewers:

PONE-D-22-13220 R1

Cocoa intake and muscle pain sensation: a randomized experimental study

Thank you for further comments and suggestions to our manuscript.

Since some questions about the statistical analysis remain, we have tried to address them in the revised text.

Please note that by reviewing the entire manuscript, some of your suggested changes (with specific lines) might have shifted. We also took the opportunity to tie up the manuscript further. 

Our point-by-point answer to your queries are presented in bold italics, ahead. 

The changes are highlighted using the track changes function in Word. We hope that these changes answer too all your queries. 

Review Comments to the Author

Please use the space provided to explain your answers to the questions above. You may also include additional comments for the author, including concerns about dual publication, research ethics, or publication ethics. (Please upload your review as an attachment if it exceeds 20,000 characters).

Reviewer #2: 

The authors have satisfactorily addressed my remaining points from the previous review, except for point 7 since the first sentence still is unclear: how could it be that findings that “the pain reduction by the different types of chocolate presented in this study is line with previous studies “ (11, 48-50); none of these 4 cited studies investigated the effects of chocolate on pain. The sentence seems unnecessary and could be deleted unless the authors modify this sentence to clarify exactly what they mean by this unclear sentence.

R: Thank you for your comment, and sorry for the confusion. We were referring to the studies that used the same procedure (e.g., pain-induced in the masseter), not chocolate. We have edited that sentence accordingly. 

Also note the revised wording on line 211 has introduced inappropriate tense changes in the same sentence; past tense should be used for both sets of verbs.

R: We are sorry for that oversight. We have corrected that sentence.

Reviewer #3: 

I think that the composition of the chocolates, now that it has been included, clearly shows that the effect, if any, is not due to the bioactives in cocoa, polyphenols and methylxanthines, but to maybe the cocoa fat and the sugar. In fact, the results show that in the case of men the sugar and fat content with no cocoa bioactives is more effective than the bioactive containing cocoa of dark chocolate.

I think that all this should be included in the abstract. It does not seem to be a question of preference but a question of sugar (and maybe fat contents)

This is in disagreement with most papers showing an effect of cocoa polyphenols and methylxanthines at different levels

See for instance:

Goya L, Kongor JE, de Pascual-Teresa S. From Cocoa to Chocolate: Effect of Processing on Flavanols and Methylxanthines and Their Mechanisms of Action. Int J Mol Sci. 2022;23(22):14365. Published 2022 Nov 18. doi:10.3390/ijms23221436

or

Gu Y, Lambert JD. Modulation of metabolic syndrome-related inflammation by cocoa. Mol Nutr Food Res. 2013;57(6):948-961. doi:10.1002/mnfr.201200837

etc

However, I think the paper should be published showing the results they got with really low amounts of cocoa solids (0% in white chocolate and only 9% in milk chocolate).

It is very important to show that this assay was done based on the bioactivity of cocoa (polyphenols and methylxanthines) however, the real content of this compounds, that was not determined, should be really low since cocoa fat has not beneficial effects that we knew until now.

R: Thank you for your suggestion and support to our findings. We have now reworded the findings in both the abstract and the text to highlight that it was the sugar content (and low amounts of cocoa solids) rather than the cocoa content alone that influenced the pain sensation.

Reviewer #4: The paper presents a randomized double-blinded controlled trial. First and foremost, the most important question if this is a powered trial to detect a significant effect. With 30 samples and without any power analysis this is not a powered trial. So any claim of significance based on p-value does not make sense.

R: Thank you for your comment, appreciated. We performed power calculation with our statistician to confirm that we had a strong power (>0.999) to detected any difference, significantly.

Second, I reviewed the previous round of reviews and agree with the statistical reviewer that this should be analyzed using a crossover design. A mixed-effect model with a period effect should be used. The previous reviewer has given some excellent guidelines on how to analyze the data which I agree with. He/she also has SAS code, which was pretty tough. Unfortunately, all the suggestions are ignored for two reasoning. 1. They do not have access to SAS and 2. The authors have used similar simpler methods earlier which are accepted.

This is not a constructive way to address the reviewer’s suggestions. First, if SAS is not available, please use R, which is completely free to use. Second, just because previous papers are not reviewed by a statistician thoroughly it does not mean the methods used by them are all appropriate. It seems the investigative teams lack a statistician as otherwise converting the given SAS code to R and reanalyzing the data using a crossover design is not a big issue. The reviewer all mentions several other challenges that have to do with normality and model fitting which I cannot comment on given the suggestion is ignored.

R: We do welcome both reviewers’ comments and suggestions. We are sorry if we came across as dismissive of your concerns, that was not the intention. We were trying to explain that SPSS did the work that was aimed to be done in our study, and we (and others) have used before. We know it is not the only way to perform statistical analysis while recognising that SPSS (point-click based) has been used in trials of the same design by many other researchers – yet, it does not make it infallible compared to SAS (programme based). We consulted with out statistician to confirm our analysis and we have also added the discussion around limitation to address this issue. We also revised the entire manuscript to better adequate the language used to present and discuss the results in light of the potential limitations. Thank you very much for understanding.

---

## [Decision Letter · Decision Letter 2]

16 Mar 2023

PONE-D-22-13220R2Chocolate intake and muscle pain sensation: a randomized experimental studyPLOS ONE

Dear Dr. Christidis,

Thank you for submitting your manuscript to PLOS ONE. After careful consideration, we feel that it has merit but does not fully meet PLOS ONE’s publication criteria as it currently stands. Therefore, we invite you to submit a revised version of the manuscript that addresses the points raised during the review process.

There are still major ponits to be addressed (see comments of reviewer 3).

We look forward to receiving your revised manuscript.

Kind regards,

Andrea Martinuzzi

Academic Editor

PLOS ONE

Reviewers' comments:

Reviewer's Responses to Questions

**Comments to the Author**

1. If the authors have adequately addressed your comments raised in a previous round of review and you feel that this manuscript is now acceptable for publication, you may indicate that here to bypass the “Comments to the Author” section, enter your conflict of interest statement in the “Confidential to Editor” section, and submit your "Accept" recommendation.

Reviewer #2: (No Response)

Reviewer #3: (No Response)

Reviewer #5: (No Response)

2. Is the manuscript technically sound, and do the data support the conclusions?

Reviewer #2: Partly

Reviewer #3: No

Reviewer #5: Partly

3. Has the statistical analysis been performed appropriately and rigorously? 

Reviewer #2: I Don't Know

Reviewer #3: Yes

Reviewer #5: No

4. Have the authors made all data underlying the findings in their manuscript fully available?

Reviewer #2: Yes

Reviewer #3: Yes

Reviewer #5: No

5. Is the manuscript presented in an intelligible fashion and written in standard English?

Reviewer #2: Yes

Reviewer #3: Yes

Reviewer #5: No

6. Review Comments to the Author

Reviewer #2: The authors have satisfactorily addressed the remaining points in my most recent review. However, I do not believe that the authors have satisfactorily addressed the point of Reviewer #3 (which was also related to a point that I raised in an earlier review) that sugar or other ingredients in the chocolate could have contributed to the reduction in pain and that this should be included in the Abstract. The revised Abstract now in its conclusion only refers to taste preference for sugar to explain the pain reduction, even though now the Discussion (lines 684-685) indicates that “other ingredients in the chocolate that could explain this chocolate-induced reduction of pain variables” and the Conclusion (lines 708-709) states that “Another possible explanation could be the composition of the chocolate, i.e. the concentration of the other ingredients such as sugar, soy, and vanilla. Comparable wording should be briefly reflected in the Abstract conclusion.

Reviewer #3: The abstract keeps being unclear and now I think that intentionally misleading. White chocolate might have 30% cocoa fat, but no flavanols at all (introduction and hypothesis is based on this). The results clearly show that if any the pleasure of fat plus sugar intake (that is what white chocolate and milk chocolate mainly provide) is the one producing the "analgesic" effect, very interestingly and in a statistically significant way in the case of white chocolate in men. I think that this needs to be said from the abstract.

Reviewer #5: The authors, in this round of revision, did their best to address the concerns raised. However, they relegated the suggestion of conducting a full-scale crossover design. It is not clear why they were reluctant to conduct it. I am OK to pass the paper, but more arguments are needed behind not using a cross-over design as a separate analysis.

7. PLOS authors have the option to publish the peer review history of their article (what does this mean?). If published, this will include your full peer review and any attached files.

Reviewer #2: No

Reviewer #3: No

Reviewer #5: No

---

## [Author Response · Author response to Decision Letter 2]

20 Mar 2023

Response to reviewers:

PONE-D-22-13220 R2

Chocolate intake and muscle pain sensation: a randomized experimental study

Thank you for further comments and suggestions to improve our manuscript, in this case the abstract.

Our point-by-point answer to your queries are presented after the R:, ahead. 

The changes are highlighted in yellow. We hope that this reply and changes in the manuscript are sufficient and will answer your queries. 

Review Comments to the Author

Please use the space provided to explain your answers to the questions above. You may also include additional comments for the author, including concerns about dual publication, research ethics, or publication ethics. (Please upload your review as an attachment if it exceeds 20,000 characters).

Reviewer #2: 

The authors have satisfactorily addressed the remaining points in my most recent review. However, I do not believe that the authors have satisfactorily addressed the point of Reviewer #3 (which was also related to a point that I raised in an earlier review) that sugar or other ingredients in the chocolate could have contributed to the reduction in pain and that this should be included in the Abstract. The revised Abstract now in its conclusion only refers to taste preference for sugar to explain the pain reduction, even though now the Discussion (lines 684-685) indicates that “other ingredients in the chocolate that could explain this chocolate-induced reduction of pain variables” and the Conclusion (lines 708-709) states that “Another possible explanation could be the composition of the chocolate, i.e. the concentration of the other ingredients such as sugar, soy, and vanilla. Comparable wording should be briefly reflected in the Abstract conclusion.

R: Thank you for your comment, and sorry for the confusion, that was not our intention. We have adjusted the conclusion making the wording comparable to the discussion and conclusion in the manuscript, as suggested. 

Reviewer #3: 

The abstract keeps being unclear and now I think that intentionally misleading. White chocolate might have 30% cocoa fat, but no flavanols at all (introduction and hypothesis is based on this). The results clearly show that if any the pleasure of fat plus sugar intake (that is what white chocolate and milk chocolate mainly provide) is the one producing the "analgesic" effect, very interestingly and in a statistically significant way in the case of white chocolate in men. I think that this needs to be said from the abstract.

R: Thank you for your comment, and sorry for the confusion, that was not our intention. We have adjusted the conclusion making the wording comparable to the discussion and conclusion in the manuscript, as suggested.

Reviewer #5: The authors, in this round of revision, did their best to address the concerns raised. However, they relegated the suggestion of conducting a full-scale crossover design. It is not clear why they were reluctant to conduct it. I am OK to pass the paper, but more arguments are needed behind not using a cross-over design as a separate analysis.

R: Thank you for your observation. We now expanded a bit further on the reasoning of not utilizing a full-blown design. We did use some of its premises, including patients being their own control, and sing a smaller sample size. However, the duration of the study might have hindered a full-scale design. We have added this point into the limitations.

---

## [Decision Letter · Decision Letter 3]

10 Apr 2023

Chocolate intake and muscle pain sensation: a randomized experimental study

PONE-D-22-13220R3

Dear Dr. Christidis,

We’re pleased to inform you that your manuscript has been judged scientifically suitable for publication and will be formally accepted for publication once it meets all outstanding technical requirements.

Kind regards,

Andrea Martinuzzi

Academic Editor

PLOS ONE

Additional Editor Comments (optional):

Reviewers' comments:

Reviewer's Responses to Questions

**Comments to the Author**

1. If the authors have adequately addressed your comments raised in a previous round of review and you feel that this manuscript is now acceptable for publication, you may indicate that here to bypass the “Comments to the Author” section, enter your conflict of interest statement in the “Confidential to Editor” section, and submit your "Accept" recommendation.

Reviewer #2: All comments have been addressed

Reviewer #3: (No Response)

Reviewer #5: All comments have been addressed

2. Is the manuscript technically sound, and do the data support the conclusions?

Reviewer #2: Yes

Reviewer #3: (No Response)

Reviewer #5: (No Response)

3. Has the statistical analysis been performed appropriately and rigorously? 

Reviewer #2: Yes

Reviewer #3: (No Response)

Reviewer #5: (No Response)

4. Have the authors made all data underlying the findings in their manuscript fully available?

Reviewer #2: Yes

Reviewer #3: (No Response)

Reviewer #5: (No Response)

5. Is the manuscript presented in an intelligible fashion and written in standard English?

Reviewer #2: Yes

Reviewer #3: (No Response)

Reviewer #5: (No Response)

6. Review Comments to the Author

Reviewer #2: (No Response)

Reviewer #3: I thank you the authors for clarifying partially the real conclusion in this study. It is their paper and if they prefer to keep the reader in the confusion that flavanols might have something to do with any of this I won't ask for rejection. However, and this is my personal recommendation, I think that the authors should know and let the readers know that there should be 0 flavanols in white chocolate, the cocoa % correspond to cocoa fat and there are no flavanols in cocoa fat. Additionally in the methodology they could give an idea of the flavanol content expected in these kind of product, that should be 0 for white chocolate and really low for the milk chocolate. I think that talking about cocoa content when the content is of cocoa fat is not rigorous.

In my opinion, in fact, the design of the study (the product chosen for the study) is not correct for the hypothesis. Maybe the authors should say so as a limitation of their study.

Reviewer #5: (No Response)

7. PLOS authors have the option to publish the peer review history of their article (what does this mean?). If published, this will include your full peer review and any attached files.

Reviewer #2: No

Reviewer #3: No

Reviewer #5: No

---

## [Editor Report · Acceptance letter]

27 Apr 2023

PONE-D-22-13220R3 

Chocolate intake and muscle pain sensation: a randomized experimental study 

Dear Dr. Christidis:

I'm pleased to inform you that your manuscript has been deemed suitable for publication in PLOS ONE. Congratulations! Your manuscript is now with our production department. 

Kind regards, 

on behalf of

Dr. Andrea Martinuzzi 

Academic Editor

PLOS ONE